# Cognitive Disengagement Syndrome (CDS) and Psychological Ill-Being in Young Adults Using the Adult Concentration Inventory (ACI)

**DOI:** 10.3390/jcm14072362

**Published:** 2025-03-29

**Authors:** Dena Sadeghi-Bahmani, Larina Eisenhut, Thorsten Mikoteit, Nico Helfenstein, Annette Beatrix Brühl, Kenneth M. Dürsteler, Stephen P. Becker, Serge Brand

**Affiliations:** 1Department of Psychology, Stanford University, Stanford, CA 94305, USA; bahmanid@stanford.edu; 2Center for Affective, Stress and Sleep Disorders, Psychiatric Hospital of the University of Basel, 4002 Basel, Switzerland; larina.eisenhut@unibas.ch (L.E.); thorsten.mikoteit@unibas.ch (T.M.); annette.bruehl@upk.ch (A.B.B.); 3Psychiatric Services Solothurn, University of Basel, 4503 Solothurn, Switzerland; 4Sport Science Section, Department of Sport and Health Science, Faculty of Medicine, University of Basel, 4052 Basel, Switzerland; nico.helfenstein@stud.unibas.ch; 5Division of Substance Use Disorders, Psychiatric University Clinics, 4002 Basel, Switzerland; kenneth.duersteler@upk.ch; 6Department for Psychiatry, Psychotherapy and Psychosomatic, Psychiatric Hospital, University of Zurich, 8057 Zurich, Switzerland; 7Division of Behavioral Medicine and Clinical Psychology, Cincinnati Children’s Hospital Medical Center, Cincinnati, OH 45229, USA; stephen.becker@cchmc.org; 8Department of Pediatrics, University of Cincinnati College of Medicine, Cincinnati, OH 45267, USA; 9Health Institute, Substance Abuse Prevention Research Center, Department of Psychiatry, Kermanshah University of Medical Sciences (KUMS), Kermanshah 6714869914, Iran; 10Sleep Disorders Research Center, Department of Psychiatry, Kermanshah University of Medical Sciences (KUMS), Kermanshah 6714869914, Iran; 11School of Medicine, Tehran University of Medical Sciences, Tehran 1417466191, Iran; 12Center for Disaster Psychiatry and Disaster Psychology, Center of Competence of Disaster Medicine of the Swiss Armed Forces, 4002 Basel, Switzerland

**Keywords:** adult concentration inventory (ACI), depression, anxiety, stress, insomnia, adolescence, sluggish cognitive tempo, cognitive disengagement syndrome (CDS)

## Abstract

**Background**: Young adulthood is a demanding developmental stage, in that individuals are often faced with making major and long-lasting decisions related to career and family. This is also a heightened time of mental health difficulties. There is recent evidence that cognitive disengagement syndrome (CDS; previously sluggish cognitive tempo) may also be more prevalent in young adults and associated with poorer functioning. However, the relation between CDS symptoms and anxiety, depression, stress, and insomnia remains insufficiently investigated among young adults. Given this, the aims of the present study were as follows: (1) to investigate the associations between CDS and symptoms of depression, anxiety, stress, and insomnia; (2) to investigate if and which dimensions of ill-being were more robustly related to higher CDS scores; (3) to explore if stress scores moderated the associations between CDS symptoms and insomnia; and (4) to explore if higher insomnia categories were associated with higher CDS scores. **Methods**: A total of 246 young adult students in Switzerland (mean age = 22.62; 56.3% females) completed a booklet of questionnaires covering socio-demographic information, cognitive disengagement syndrome (Adult Concentration Inventory; ACI), and symptoms of depression, anxiety, stress, and insomnia as part of this cross-sectional study. **Results**: Higher CDS scores on the ACI were associated with higher scores for depression, anxiety, stress, and insomnia. Depression, anxiety, stress, and insomnia were independently associated with higher scores for CDS. Higher categories of stress moderated the associations between higher CDS scores and higher insomnia. Higher insomnia categories were related to higher CDS scores. **Conclusions**: The present data showed that among a small sample of young adult students, higher CDS scores were associated with higher psychological ill-being (depression, anxiety, stress, and insomnia). If we consider CDS as a trait, specific performance-enhancing medication or psychotherapeutic interventions might favorably influence dimensions of psychological ill-being such as depression, anxiety, stress, and insomnia.

## 1. Introduction

As a general observation and compared to adolescence and middle and late adulthood, early adulthood is a particularly challenging developmental period of life, as individuals in this age range have to cope with many vocational and psychosocial developmental tasks, including vocational, social, and emotional tasks, during a unique transition period between adolescence and adulthood [1].

Further, several lines of research showed that young adult students are at increased risk to suffer from mental health issues [2,3,4,5,6,7,8].

Given that vocational and academic performance is one of the key issues in early adulthood, plausibly, questions arise as to why some young adults are more successful or less successful in achieving their academic or vocational goals. Unsurprisingly, one key point is the capacity to stay cognitively persistently focused on academic/intellectual tasks [9]. A further observation is that adolescent and adult individuals suffering, for instance, from attention-deficit/hyperactivity (ADHD) report lower school and academic grades, higher rates of school break-ups, more failures in vocational trainings, more social issues at the workplace, and lower incomes than their non-ADHD counterparts [10,11,12].

### 1.1. Cognitive Disengagement Syndrome

Besides the cognitive performance–ADHD link, in recent years, a growing body of research has focused on a similar, though clinically distinguishable cognitive behavioral set labeled cognitive disengagement syndrome (CDS; formerly known as sluggish cognitive tempo [SCT]).

Cognitive disengagement syndrome (CDS) refers to a set of attentional symptoms that includes daydreaming, staring, mental fogginess/confusion, slowed behavior/thinking, and hypoactivity [13,14]. Cross-sectional, longitudinal, and meta-analytic studies consistently report the robust pattern of CDS being distinguished from inattentive symptoms of the attention-deficit/hyperactivity disorder (ADHD-IN) [15]. Importantly, among children and adolescents, such a pattern of CDS could be observed among different cultural areas and languages such as American English [16,17,18,19], Spanish [20,21], Korean [22], Turkish [23,24], and Persian/Farsi [25].

While it appears that the concept of CDS is well-established among children and adolescents, this is less the case for adults. In an important study examining CDS in a sample of 1249 adults aged 18 to 96 years, those participants identified with CDS (n = 33; CDS + ADHD: n = 39; 5.76% of the sample) had a lower annual income, reported more problems with self-organization and problem solving, had lower levels of work and education, were rather out of work, and had more social impairments. Sex, age, and ethnicity were not confounders [14]. Importantly, the CDS link to emotional issues (i.e., symptoms of depression, anxiety, and stress) and insomnia has been poorly investigated so far, and CDS scores appeared to be much more common among adults seeking an ADHD evaluation [26].

### 1.2. Adult Symptoms of Cognitive Disengagement Syndrome (CDS) in Relation to Cognitive, Attentional, Emotional, Behavioral, and Sleep-Related Difficulties

For the CDS–cognitive performance link, data from 60 cross-sectional and longitudinal studies showed that higher scores for CDS were associated with poorer academic performance [27]. More specifically, higher CDS symptoms were associated with the following: 1. poorer skills in daily life, executive functioning, and greater functional impairment in specific domains of educational activities, work (see also [14]), money/financial issues, managing chores and household tasks, community activities, and social situations with strangers and friends [28]; 2. lower scores for self-organization and problem solving [29]; 3. higher scores for a distorted time perception [30]; 4. a weaker orienting network due to the problems of engaging and disengaging the attention [31]; 5. a reduced speed and efficacy of selective attention in early information processing [32]; 6. more deficits in the use of self-regulated learning strategies [33]; and 7. more difficulties in a timed reading test, though students with CDS were not slower than controls on the reading comprehension, processing speed, and reading fluency [34]. By contrast, more recent research provided clearer evidence for a link between CDS and a wide range of academic [35] and neurocognitive [36] outcomes.

As regards studies on CDS focusing on emotional difficulties, including stress and social behavior, adults scoring high on CDS were at increased risk to report symptoms of suicidal behavior, beyond other health dimensions, including symptoms of depression [3]. Among college students aged 17 to 34 years, higher scores for CDS were associated with more academic impairments, anxiety, and depression, even after controlling for sex, age, ethnicity, parent education, family income, and ADHD-related medication [37]. Among 158 young adults (mean age: 19.05 years), higher CDS scores were associated with higher scores for depression and anxiety and lower scores for emotion regulation and social adjustment [38]; above and beyond other psychopathologies, CDS was significantly associated with social impairments, with emotion regulation as a moderator. Among 274 outpatients aged 18 to 64 years with a broad variety of diagnosed psychiatric disorders, CDS scores showed stronger first-order and unique associations than ADHD-IN with symptoms of depression, anxiety, and stress, along with sleep disorders [39].

Next, higher scores for ADHD predicted higher scores for self-perceived stress, while the combination of symptoms of ADHD inattentive type (ADHD-IN) and symptoms of CDS was the most consistent predictor of perceived stress [40]. Higher scores for emotion dysregulation moderated the associations between higher scores for CDS and social impairment. Importantly, CDS traits and social withdrawal appeared to be highly intertwined [38]. To explain such an association, the conceptual framework considers task-unrelated thoughts, poorer social skills, and social anxiety, along with possible moderators such as behavioral inhibition and unfavorable parenting styles, in the emergence and maintenance of the CDS and social withdrawal link [41]. Among 274 young adults aged 18–35 years (70.4% females), higher scores for CDS were associated with higher risks for internet gaming disorder and internet addiction [42]. If we consider addictive behavior as a means to regulate emotions [43,44,45], this assumption appears to match the concept that higher CDS scores and lower stress coping strategies are associated. Kamradt et al. [46] calculated among 158 young students that CDS scores moderated the association between mental health comorbidities and cognitive dysfunctions, including internalizing problems. Accordingly, Kamradt et al. [46] claimed that CDS scores should be further considered as a transdiagnostic link between psychological ill-being and cognitive dysfunctions. Among a larger sample of 1024 adults (62.5% females), those scoring high on CDS and ADHD showed more difficulties in emotion regulation, including higher scores for alexithymia. However, in the regression model, only CDS scores, but not ADHD scores, were significantly associated with poor emotion regulation and higher alexithymia [47].

As regards the associations between CDS scores and sleep parameters, higher scores for CDS were associated with more impaired sleep patterns [48,49] and with a lower sleep quality, along with more daytime dysfunctioning [50], with more sleep disturbances, along with higher scores for symptoms of stress, depression, and functional impairment [51]. Importantly, symptoms of inattention and CDS predicted daytime dysfunctioning above and beyond sleep quality [37]. Among 274 outpatients aged 18 to 64 years with a broad variety of diagnosed psychiatric disorders, CDS scores showed stronger first-order and unique associations than ADHD-IN with sleep problems, along with symptoms of depression, anxiety, and stress [39]. Lunsford-Avery et al. [52] assessed 65 adults from an outpatient medical center aged 19 to 63 years. Of those, 43 had a diagnosis of ADHD, and 22 had internalizing and adjustment disorders. Higher eveningness scores were associated with higher CDS scores, and the combination of eveningness and CDS scores was strongly associated with internalizing and adjustment disorders.

### 1.3. The Present Study

Young adults are particularly under pressure to perform cognitively, emotionally, and psychosocially and as regards their academic and vocational track; though, CDS interferes with demands to perform, which might further increase psychological ill-being such as depression, anxiety, stress, and insomnia. However, such data were scarcely sampled concomitantly and above all among non-US university students. To counter this, we assessed a sample of Swiss university students, who completed a series of self-rating questionnaires on CDS, depression, anxiety, stress, and insomnia.

The following four hypotheses and study questions were formulated. First, following others [37,38,39,40,53,54], we expected that higher scores for CDS were associated with higher scores for depression, anxiety, and stress. Second, following previous studies [39,48,49,51,55,56], we hypothesized that higher scores for CDS were associated with higher insomnia scores. Third, following a previous study [57], we assumed that stress moderated the associations between CDS scores and insomnia. Last, again following others [39,48,49,51,55,56], we expected that higher insomnia categories were associated with higher CDS scores.

We believed that the present data have the potential to add to the current CDS literature in the following important ways: First, we further substantiated the CDS–psychological ill-being link, though among a non-clinical sample of young adults in a country where CDS has yet to be examined. Second, we specifically investigated the importance of insomnia to understand the CDS–psychological ill-being link, and third, we explored if and to what extent self-perceived stress might moderate the associations between CDS and insomnia. To this end, young adult students completed a series of self-rating questionnaires covering CDS, depression, anxiety, stress, and insomnia.

## 2. Materials and Methods

### 2.1. Procedure

Similar to a previous study [58], students of the University of Basel (Basel, Switzerland) were invited to participate in the present cross-sectional and anonymous online study, which was performed with the online survey software Tivian^®^/Questback^®^. The manufacturer warrants that all data are securely stored on the manufacturer’s server, that no third parties have access to the data, and that no hidden information of a participant, such as the IP address, will be gathered and stored.

On the first page of the online study, participants were fully informed about the aims of this study, the anonymous data gathering, and data elaboration. Participants were also informed that participation or non-participation had no advantages or disadvantages for the continuation of this study and that a participant could stop or interrupt the participation at any time. The first page of the online study further indicated that the present data might be used for scientific research and publication. Next, to ‘sign’ the written informed consent, participants were asked to tick the following box: “I have understood the aims of this study, including the anonymous data gathering and the secure and anonymous data handling. I know that I can withdraw from this study without further consequences, and I know whom to contact in case of further study-related questions”. Afterwards, participants completed online questionnaires covering sociodemographic information, symptoms of CDS, and symptoms of psychological ill-being, that is, depression, anxiety, stress, and insomnia. On average, participants needed about 30–35 min to complete the online questionnaires (see details below). This study lasted from 11 November 2024 to 31 December 2024. The local ethical committee (Ethikkommission Nordwest- und Zentralschweiz [EKNZ], Basel, Switzerland) approved this study (Register No.: Req-2024-01463; approved on 11 November 2024), which was performed according to the seventh [59] and current version of the Declaration of Helsinki.

### 2.2. Participants

Inclusion criteria were as follows: 1. age between 18 and 30 years; 2. student at the University of Basel (Basel, Switzerland); 3. willing and able to comply with the study requirements, including the intermediate to mastery level in German; and 4. ticking the box to “sign” the written informed consent. Exclusion criteria were as follows: 1. withdrawing from participation; 2. pregnancy or breast-feeding, as such stages may alter both mood and sleep; and 3. ‘click-throughs’ who needed less than five minutes to complete the questionnaires.

A total of 746 individuals read the first page, 512 (68.63%) started the online survey, and 246 (32.98%) completed it. Of those, one (0.13%) self-declared to be gender diverse; given this very low prevalence rate of self-declared individuals being diverse, data of this person were not further considered for data analysis. The full dataset consisted of 245 participants (32.84%; mean age: 22.62 years (SD = 3.10); 56.3% females).

### 2.3. Measures

#### 2.3.1. Sociodemographic Information

Participants reported on their gender (male, female, and diverse) and age (in years).

#### 2.3.2. Adult Concentration Inventory (ACI)

The ACI is a self-report measure to assess CDS symptoms [60,61,62]. Example items are “I’m slow at doing things”, “I get lost or drowsy during the day”, or “I get lost in my own thoughts”. Each item is rated on a four-point scale (0 = not at all, 1 = sometimes, 2 = often, and 3 = very often), with a higher sum score reflecting greater CDS symptoms severity (Cronbach’s alpha: 0.94). As in previous studies [39,62], the 15-item version, and not the 16-item version, was used, as in previous validations the item “I’m not motivated” turned out to be a non-optimal CDS item.

To accurately translate the English version of the ACI into German, we followed the algorithms proposed by Brislin [63], Beaton et al. [64], and Sousa and Rojjanasrirat [65] (see also Sadeghi-Bahmani et al. [25]: (i) two independent translators with expertise in both German and English translated the English version of the ACI into German; (ii) a third independent person with expertise in both German and English compared the two translations; (iii) where there were differences, the three experts discussed the issues and formulated the final draft; (iv) two further and new independent translators with expertise in both German and English performed the back-translation, and (v) compared the back-translated English version with the original version; and (vi) the final version reflected the general agreement of all five researchers involved in this procedure. The German version of the Adult Concentration Inventory (ACI) is available in Appendix A Materials (see the end of the present file).

#### 2.3.3. Depression, Anxiety, and Stress

To assess symptoms of depression, anxiety, and stress, participants completed the Depression, Anxiety, Stress-questionnaire [DASS-21; [66,67]]. The questionnaire consists of 21 items, and example items are “I felt down and depressed” [depression]; “I felt I was close to panic” [anxiety]; and “I was in a state of nervous tension” [stress]. Answers are given on 4-point Likert scales ranging from 0 (=does not apply to me at all) to 3 (=extremely applies to me), with higher mean scores reflecting a higher severity of symptoms. Accordingly, the subscale depression, the subscale anxiety, and the subscale stress were calculated separately (Cronbach’s alphas of the overall score = 0.96).

#### 2.3.4. Insomnia

To assess insomnia, participants completed the German version [68] f the Insomnia Severity Index (ISI; [69]). It includes seven questions about sleep quality and insomnia; participants answered how often certain conditions concerning sleep quality have occurred during the last month on scales ranging from 0 (=never/not at all) to 4 (=always). The total score ranges from 0 to 28, with a higher sum score reflecting a higher severity of insomnia (Cronbach’s alpha: 0.88). The following cut-off values are proposed [69]: 0–7 points: no insomnia; 8–14 points: subthreshold insomnia; 15–21 points: moderately clinically relevant insomnia; and 22–28 points: severely clinically relevant insomnia.

### 2.4. Analytic Plan

#### Preliminary Calculations

With a series of Pearson’s correlations, we investigated if age was systematically associated with CDS and psychological ill-being. All rs were <0.12 (*p*s > 0.7). Thus, age was not introduced as a confounder.

With a series of *t*-tests, we investigated if male and female participants did systematically differ in age and in CDS and psychological ill-being. Compared to males (M = 21.90; SD = 2.79), females were older (M = 23.19; SD = 3.22; t(243) = 3.30, *p* < 0.01, d = 0.43); females reported higher anxiety scores (males: M = 1.84 (SD = 2.45); females: M = 2.71 (SD = 3.57); t(243) = 2.15, *p* < 0.05, d = 0.28) and CDS scores (males: M = 8.91 (SD = 7.52); females: M = 11.40 (SD = 9.32; t(243= = 2.25, *p* < 0.05, d = 0.29). Given the small effect sizes, the decision was not to introduce gender as a confounder.

With a series of Pearson’s correlations, we explored the associations between scores for CDS and depression, anxiety, stress, and insomnia.

To explore which dimensions of psychological ill-being (depression, anxiety, stress, and insomnia) were more strongly associated with CDS scores, a multiple regression analysis was performed. The following statistical requirements for running a multiple regression model were met [70,71,72]: N = 245 > 100; predictors explained the dependent variable (R = 0.824; R^2^ = 0.679); number of predictors: 4 (depression, anxiety, stress, and insomnia): 10 × 4 = 40 < N (245); and the Durbin–Watson coefficient was 1.59, indicating that the residuals of the predictors were independent. Last, the variance inflation factors (VIF) were between 2.15 and 4.49; while there are no strict cut-off points to report the risk of multicollinearity, a VIF < 1 and a VIF > 10 indicate multicollinearity [70,71,72].

Next, to test whether perceived stress moderated the association between CDS scores and insomnia, we followed Aiken and West [73]. More specifically, Aiken and West [73] suggested splitting a possible moderator into a categorical variable. Accordingly, we split perceived stress into the following three categories: low: m = 1.06, n = 82; medium: m = 4.75, n = 100; and high: m = 11.00, n = 63, and we introduced these three categories as moderators between CDS and insomnia (see also [57]).

Last, to test whether categories of insomnia did systematically change CDS scores, an ANOVA was performed with insomnia categories as the factor and CDS scores as the dependent variable. Post hoc tests were performed after Games–Howell.

The level of significance was set at alpha < 0.05. All statistical computations were performed with SPSS^®^ 29.00 (IBM Corporation, Armonk, NY, USA) for Apple Mac^®^.

## 3. Results

### 3.1. General Information

Full data were available for 245 participants. Their mean age was 22.62 years (SD = 3.10); 107 (43.7%) were males, and 138 (56.3%) were females.

### 3.2. Associations Between Cognitive Disengagement Syndrome and Symptoms of Depression, Anxiety, Stress, and Insomnia

Table 1 reports the descriptive statistical indices of CDS and symptoms of depression, anxiety, stress, and insomnia and the correlation coefficients (Pearson’s correlations).

Higher scores for CDS were associated with higher scores for depression, anxiety, stress, and insomnia.

Higher scores for depression were associated with higher scores for anxiety, stress, and insomnia.

Higher scores for anxiety were associated with higher scores for stress and insomnia.

Higher scores for stress were associated with higher scores for insomnia.

### 3.3. Regression Model to Identify Which Dimensions of Ill-Being (Depression, Anxiety, Stress, and Insomnia) Were More Robustly Associated with Scores for Cognitive Disengagement Syndrome (CDS)

Table 2 reports the regression model with CDS scores as the outcome variable and depression, anxiety, stress, and insomnia as predictors.

It turned out that higher scores for depression, anxiety, stress, and insomnia were independently associated with higher CDS scores.

### 3.4. Categories of Stress Moderators for the Association Between CDS Scores and Insomnia Scores

As shown in Table 1, scores for CDS, stress, and insomnia were highly intertwined. We asked whether categories of stress moderated the association between CDS and insomnia scores. To this end, we followed Aiken and West [73] and split stress scores into the following three categories: low: m = 1.06, n = 82; medium: m = 4.75, n = 100; and high: m = 11.00, n = 63.

While the overall Pearson’s correlation coefficient between CDS and insomnia scores was r = 0.69 *** (see Table 1), the correlation coefficient changed to r = 0.36 *** in the low stress condition, to r = 0.37 *** in the medium stress condition, and to r = 0.64 *** in the high stress condition. Figure 1 shows this pattern.

### 3.5. Categories of Insomnia and Cognitive Disengagement Syndrome Scores

To further explore if and to what extent CDS scores systematically changed as a function of insomnia (r = 0.69 ***; see Table 1), insomnia scores were split into the following cut-off values: 0–7 points: no insomnia; 8–14 points: subthreshold insomnia: 15–21 points: moderately clinically relevant insomnia; and 22–28 points: severely clinically relevant insomnia.

Table 3 reports the Insomnia Severity Index categories (n, m, and SD), the related CDS scores, and the inferential statistical indices of the ANOVA.

Higher Insomnia Severity Index categories had statistically significantly higher CDS scores.

## 4. Discussion

The aims of the present study were to investigate among a sample of young adult students the associations between cognitive disengagement syndrome (CDS) scores and dimensions of psychological ill-being, namely depression, anxiety, stress, and insomnia. The key findings were as follows: (1) higher CDS scores were associated with higher scores for depression, anxiety, stress, and insomnia; (2) in the regression model, depression, anxiety, stress, and insomnia were independently associated with higher CDS scores; (3) higher stress categories moderated the association between CDS scores and insomnia; and (4) higher insomnia categories were associated with higher CDS scores. The present pattern of results adds to the current literature in the following five ways. First, this is the first study investigating the associations between CDS scores and psychological ill-being among a non-US and non-clinical sample of students in their young adulthood. Second, we concomitantly assessed symptoms of depression, anxiety, stress, and insomnia, and third, we showed that symptoms of depression, anxiety, stress and insomnia independently were significantly and strongly associated with higher CDS scores. Fourth, we observed that stress moderated the CDS–insomnia link, and fifth, we further substantiated the tight insomnia–CDS link.

Four hypotheses were formulated, and every single one was considered.

### 4.1. CDS Scores and Psychological Ill-Being

With the first hypothesis we assumed that higher scores for CDS would be associated with higher scores for psychological ill-being, namely depression, anxiety, and stress, and data did confirm this (see Table 1). As such, we also confirmed previous findings [37,38,39,40,53,54]. However, we expanded upon previous studies in that we observed such a pattern of results among a non-US American and non-clinical sample of young adults. Further, indirectly, we confirmed that CDS might be considered a transdiagnostic psychiatric dimension [46], and we also showcased that CDS is associated with internalizing problems.

### 4.2. CDS Scores and Insomnia

With the second hypothesis we assumed that higher scores for CDS would be associated with higher scores for insomnia, and data did confirm this. As such, we mirrored previous findings [39,48,49,51,55,56]. The plus of the present study was that unlike the majority of previous studies, we assessed insomnia concomitantly with dimensions of psychological ill-being (i.e., depression, anxiety, and stress), that insomnia was a main outcome variable, and that we used a psychometrically validated measure (ISI; Insomnia Severity Index [69]), which allowed us to report insomnia both as a continuous and as a categorical dimension (see also the discussion of the fourth hypothesis).

### 4.3. CDS, Insomnia, and Stress

With the third hypothesis we assumed that stress would moderate the association between CDS scores and insomnia, and again, data did confirm the following: higher stress categories described the CDS–insomnia link more strongly (see Figure 1). We have taken this kind of pattern from a previous study, where stress moderated the association between insomnia and dark triad traits in the military context [57]. As such, while previous studies reported either a CDS–stress link [37,38,39,40,53,54] or a CDS–insomnia link [39,48,49,51,55,56], the beauty of the present study lies in combining a CDS–insomnia–stress link, which allowed a more fine-grained data analysis.

### 4.4. CDS and Insomnia Categories

With the fourth and last hypothesis, we assumed that more severe insomnia categories predicted higher CDS scores, and data again did confirm this assumption.

### 4.5. Transdiagnostic and Allostatic Models to Explain the CDS–Psychological Ill-Being–Insomnia Link

To explain the CDS–psychological ill-being–insomnia link, we consider two models.

Two psychiatric and cognitive–behavioral concepts help to explain why the effective treatment of a specific psychiatric disorder leads to improvements in other psychiatric conditions (see also [74]). For instance, treating symptoms of insomnia also improved symptoms of depression [75], anxiety, and stress [76,77,78,79,80]. To explain this phenomenon, the concept of the transdiagnostic approach [79,81,82,83,84,85,86,87] reflects the observation that improvements in one dimension of psychological experiences are associated with improvements in further dimensions of psychological experiences. Moreover, a meta-analysis on the treatment of anxiety disorders did not observe systematic differences between a disorder-specific cognitive–behavioral therapy (CBT) and a transdiagnostic CBT (tCBT) [87]. Further, no associations between the comorbidity rate and tCBT outcome were observed. 

Next, the concept of allostatic load [88,89,90] may help explain the improvements of non-specific benefits of CBT. Allostatic load is understood as the cumulative effects of stressful experiences in daily life, and such an allostatic load may lead to both physiological and psychological strain over time. Thus, almost by default, it is conceivable that higher scores for CDS as operationalized with the Adult Concentration Inventory [61,62] might be associated with higher scores for depression, anxiety, stress, and insomnia.

### 4.6. Limitations and Future Directions

Limitations of this study were as follows: First, the sample itself was highly selected (young adult students not older than 30 years), which precludes transferring the present results to the general adult population. Second, the cross-sectional study design precludes understanding if, and if so, to what extent CDS scores were the cause or the result of psychological ill-being. However, there is some evidence that CDS predicts subsequent depressive symptoms [91,92], and future longitudinal (e.g., see [93,94,95,96,97]) and interventional [98,99] study designs should help to understand the causal directions. Third, it is conceivable that further latent and unassessed psychological dimensions such as ADHD(-IN), regular physical activity, medication intake, study exams, socioeconomic issues [100,101], or daily hassles may have biased two or more dimensions in the same or opposite directions. In this view, consider that about 30% of students report psychological health issues [2,3,4,5,6,7,8,51,102,103,104,105,106,107,108,109,110,111,112,113,114], including medical students [58]. Thus, future studies should include larger samples with broader age ranges of the general population.

## 5. Conclusions

Among a sample of young adults, higher scores for CDS were associated with higher scores for depression, anxiety, stress, and also insomnia. Depression, anxiety, stress, and insomnia were independently associated with higher CDS scores, and stress moderated the association between CDS and insomnia. If we consider CDS as a trait, specific performance-enhancing medication or psychotherapeutic interventions might favorably influence dimensions of psychological ill-being such as depression, anxiety, stress, and insomnia.

## Figures and Tables

**Figure 1 jcm-14-02362-f001:**
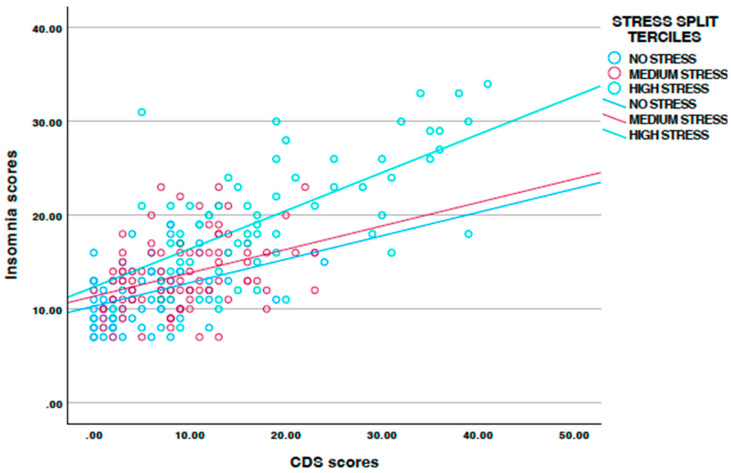
Associations between CDS and insomnia scores, moderated by stress categories.

**Table 1 jcm-14-02362-t001:** Descriptive statistical indices of and correlation coefficients (Pearson’s correlations) between scores for cognitive disengagement syndrome, depression, anxiety, stress, and insomnia (N = 245).

	Dimensions				
	CDS	Depression	Anxiety	Stress	Insomnia
CDS	-	0.77 ***	0.74 ***	0.75 ***	0.69 ***
Depression		-	0.83 ***	0.81 ***	0.72 ***
Anxiety			-	0.78 ***	0.62 ***
Stress				-	0.67 ***
Insomnia					-
M (SD)	10.31 (8.66)	3.33 (4.42)	2.33 (3.17	5.12 (4.31	7.64 (5.57)

Notes: CDS = cognitive disengagement syndrome; *** = *p* < 0.001.

**Table 2 jcm-14-02362-t002:** Multiple linear regression with cognitive disengagement syndrome scores as the outcome variable and depression, anxiety, stress, and insomnia as predictors.

Dimension	Variables	Coefficient	Standard Error	Coefficient β	t	*p*	R	R^2^	Durbin–Watson	VIF
CDS	Intercept	2.176	0.595	-	3.656	<0.001	0.824	0.673	1.59	
	Depression	0.482	0.153	0.246	3.160	0.002				4.494
	Anxiety	0.550	0.190	0.201	2.896	0.004				3.582
	Stress	0.480	0.136	0.239	3.541	<0.001				3.398
	Insomnia	0.364	0.084	0.233	4.343	<0.001				2.149

Notes: CDS = cognitive disengagement syndrome.

**Table 3 jcm-14-02362-t003:** Cognitive disengagement syndrome scores, split by Insomnia Severity Index categories.

	Dimensions			Statistics	
		Insomnia Severity Index Scores	CDS Scores	ANOVA for CDS Scores	Post Hoc Tests After Games–Howell for CDS Scores
Insomnia Categories	n	M (SD)	M (SD)		
No insomnia	11	7 (0.00)	4.63 (4.43)	F(3, 241) = 67.11, *p* < 0.001; partial eta squared: 0.455	No insomnia = subthreshold insomnia No insomnia < moderate clinical insomnia and severe clinical insomnia
Subthreshold insomnia	135	11.29 (1.82)	6.55 (4.98)		Subthreshold insomnia = no insomnia Subthreshold insomnia < moderate clinical insomnia and severe clinical insomnia
Moderate clinical insomnia	73	17.56 (1.94)	12.93 (7.14)		Moderate clinical insomnia > no insomnia and subthreshold insomnia Moderate clinical insomnia < severe clinical insomnia
Severe clinical insomnia	26	26.62 (3.72)	24.92 (10.52)		Severe clinical insomnia > no insomnia, subthreshold insomnia, and moderate clinical insomnia

Notes: CDS = cognitive disengagement syndrome.

## Data Availability

Data might be made available under the following conditions: 1. Only an internationally recognized senior researcher can ask for the data set. 2. The scientific profile of the senior researcher is easily retrievable on the homepage of the institution. 3. The senior researcher contacts the corresponding author via her/his institutional email address (no @gmail.com or similar). 4. The senior researcher formulates clear-cut hypotheses; such hypotheses transparently describe the reasons as to why the data set should be provided. 5. The senior researcher describes credibly how the data set is securely stored on an institutional server, which is not accessible to a third party. 6. The senior researcher declares that the data set by no means is shared with a third party.

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
