# Peer review of "Cognitive Disengagement Syndrome (CDS) and Psychological Ill-Being in Young Adults Using the Adult Concentration Inventory (ACI)"

_jcm, 2025, doi:10.3390/jcm14072362_

Round 1

Reviewer 1 Report

Comments and Suggestions for Authors
  • I am grateful for the opportunity to review this work. The following represent the most important opportunities I see for strengthening this manuscript: The topic sounds interesting but the paper is not well organized and the study has minor defect. There are some point need further explain, For example, it is confused about the rationales for this study.
  • Abstract the structure should still be in line with an empirical research format. The abstract should be revisited. Rephrasing the research summary. Especially the conclusion part.
  • The objectives appear broad. The second objective  should explicitly indicate why this was hypothesized or investigated. The rationale behind the focus on gender differences needs elaboration.
  • What is novelty in this study? There are so many studies conducted on this domain. The existence of many studies that dealt with the same topic.
  • In what ways does this research benefit both society as whole and existing fields of study? Use how this research can in future.
  •   I hope this helps.

Author Response

We thank Reviewer #1 for the kind efforts devoted to the present manuscript. Please see the detailed point-by-point-response attached as a separate file. 

Thank you again for all your efforts. 

Reviewer 2 Report

Comments and Suggestions for Authors

I read the manuscript jcm-3478579 entitled "Associations between Cognitive Disengagement Syndrome (CDS) and Symptoms of Depression, Anxiety, Stress, and Insomnia in Young Adult Students Using the Adult Concentration Inventory (ACI)." I enjoyed reading the paper; it introduces an important topic in the domain of mental well-being research. Authors gathered data from 246 young adult students in Switzerland and found that higher cognitive disengagement syndrome (CDS) scores on the ACI were associated with greater levels of depression, anxiety, stress, and insomnia.

I recommend making some minor revisions to improve the quality of the manuscript.

  1. The topic is too wordy; I suggest authors keep it concise, using a maximum of 12 words.
  2. There is a gap in the research that needs to be addressed. While CDS is a significant contributor to mental health disorders, several other factors—such as demographic, socioeconomic, and geographical elements—also play a crucial role. These factors include family income, financial liabilities, income shocks, unemployment, living location, and religion, among others, which can impact a wide range of mental health issues, particularly anxiety and insomnia. I recommend that authors emphasize this point in the introduction. There are several papers that can help close the gap, listed as follows:

Black, N.Jackson, A., & Johnston, D. W. (2022). Whose mental health declines during economic downturns? Health Economics18.

Archer, J., & Rhodes, V. (1993). The grief process and job loss: A cross‐sectional study. British Journal of Psychology84(3), 395-410.

Zamanzadeh, A., Cavoli, T., Ghasemi, M., & Rokni, L. (2024). The effect of actual and expected income shocks on mental wellbeing: Evidence from three East Asian countries during COVID-19. Economics & Human Biology53, 101378.

           3.  Results section should be elaborated, particularly concerning Table 2. Where is the model? What is the model specification? Where are the estimates of mental health effects of other control variables such as age, gender, and others? This issue should be clarified to show authors are aware of omitted variables bias, even if estimates of control variables are not provided in the paper.

         4. Conclusion is too short. I suggest the authors discuss future research and potential gaps that can be addressed. 

Author Response

We thank Reviewer #2 for the kind efforts devoted to the present manuscript. Please see the detailed point-by-point-response attached as a separate file. 

Thank you again for all your efforts. 
